# Psychometrics of the Pearlin Mastery Scale among Family Caregivers of Older Adults Who Require Assistance in Activities of Daily Living

**DOI:** 10.3390/ijerph19084639

**Published:** 2022-04-12

**Authors:** Zhi Xiang Lim, Wei Ling Chua, Wee Shiong Lim, An Qi Lim, Kia Chong Chua, Ee-Yuee Chan

**Affiliations:** 1Nursing Research Unit, Nursing Service, Tan Tock Seng Hospital, Singapore 308433, Singapore; melimzx@hotmail.com (Z.X.L.); an_qi_lim@nuhs.edu.sg (A.Q.L.); 2Alice Lee Centre of Nursing Studies, National University of Singapore, Singapore 119077, Singapore; chuaweiling@nus.edu.sg; 3Department of Geriatric Medicine, Institute of Geriatrics and Active Aging, Tan Tock Seng Hospital, Singapore 308433, Singapore; wee_shiong_lim@ttsh.com.sg; 4Institute of Psychiatry, Psychology & Neuroscience, King’s College London, London SE5 8AF, UK; kia-chong.chua@kcl.ac.uk

**Keywords:** caregivers, item response theory, confirmatory factor analysis, mastery, older adults, scale, psychometric, reliability, validity

## Abstract

This study examined the psychometric properties of the seven-item mastery scale among 392 family caregivers of care dependent older adults in a tertiary hospital in Singapore. Item response theory (IRT) analysis and confirmatory factor analysis (CFA) were used to assess the scale’s psychometric properties. Construct validity was assessed based on correlations between mastery and caregiver burden, depression, and quality of life. Data from the seven-item mastery scale showed acceptable reliability and model fit while IRT analysis showed that response categories were ordered but reflected poor fit for the two positively worded items. Without these two items, responses on the five-item version showed acceptable model fit and had acceptable reliability and high correlation with those on the seven-item version. Item responses on both the seven- and five-item versions show logical correlations with carer self-report on burden, depression, and quality of life. Further psychometric studies of the seven-item mastery scale are warranted. For practical applications such as caregiver screening during hospital admissions, the five-item mastery scale is fit for purpose.

## 1. Introduction

Mastery is the degree to which individuals feel that they have control over their life situations [1]. Unlike constructs such as self-efficacy, mastery is a more global concept and does not apply to specific situations or tasks. In the original works of Pearlin, Mullan, Semple and Skaff [2,3] amongst caregivers of person with dementia, mastery was conceptualized as a self-concept that can buffer against stressful events and psychological distress. Hence, high levels of mastery amongst caregivers are important given that caregiving is often stressful and can be overwhelming [4].

The protective effects of mastery have been documented in several populations. In caregivers of older adults who require assistance with daily living such as dementia patients, mastery has been shown to be negatively associated with caregiver burden, anxiety and depression, and positively associated with health-related quality of life (HRQoL) [4,5]. This suggests that having mastery may buffer the negative health effects due to stress associated with caregiving [4,5]. Similarly, studies on people with mental illness found that mastery mediates the relationship between psychopathology and perceived health [6]. Among the unemployed, those with high mastery had lower depressive symptoms than those who had low mastery [7]. Importantly, the malleability of mastery was also highlighted in interventional studies whereby the mastery levels of caregivers increased after undergoing psychotherapeutic interventions [8].

Pearlin and Schooler’s personal mastery scale [1] is the most widely used scale to measure mastery. The original seven-item, one factor version comprises five negatively and two positively worded items (Appendix A). To date, few studies have examined the psychometric properties of the mastery scale. Those which did so had largely focused on the psychometric analyses of the translated versions of the original scale into Chinese, Swedish, Spanish, and Japanese languages among individuals from the patient or community-dwelling population [9,10,11,12].

While past studies have found that the mastery scale has acceptable psychometric properties, a common finding was the poor performance of the two positively worded items, (item no. 4 and 6). In particular, a Japanese study conducted with residents in the community, showed that removing the two positively worded items would result in better reliability and confirmatory factor analysis (CFA) fit statistics [9]. This was supported by another Taiwanese study conducted with patients from the outpatient medical center and municipal hospital, which called for the removal of these two positively worded items which displayed poor factor loadings in CFA [12]. Additionally, Canadian studies have also used a five-item version of the mastery scale for analysis as the seven-item version proved to have poor reliability statistics in their community sample of residents [13,14].

Specifically, while the mastery scale has been frequently used in research involving caregivers of care dependent older adults [2,3,15,16,17,18,19,20], no study has done a thorough investigation of the scale in this population. Thus, this study aims to evaluate the psychometric properties of the mastery scale among family caregivers, applying the item response theory (IRT) [21]. IRT discrimination and difficulty parameters were used to understand the performance of each item in the mastery scale [21,22]. If the two positively worded items (items no. 4 and 6) of the mastery scale are shown to perform poorly in the IRT analysis, we also aimed to assess the reliability and validity of the five-item version by removing these two items.

## 2. Methods

### 2.1. Participants and Procedure

The current sample combines two cohorts of family caregivers of hospitalized older adults from a 1700 bed tertiary hospital in Singapore [4,15]. In both samples, family caregivers completed a questionnaire consisting of the seven-item mastery scale after patients’ hospitalization, once patients were medically stable. This included family caregivers of patients who fulfilled the following criteria: (1) aged 65 years and above, (2) requiring assistance in activities of daily living, and (3) residing in the community. The inclusion criteria were determined via patients’ medical records or by asking the nurse-in-charge of the patient. Participants who met the criteria completed the questionnaires in English or Chinese, depending on which language they were more comfortable with. Chinese versions of the scales used were previously validated [23,24].

### 2.2. Measures

#### 2.2.1. Mastery

The seven-item personal mastery scale developed by Pearlin and Schooler [1] was used to measure mastery. All items are rated on a four-point scale: 1 = strongly agree; 2 = agree; 3 = disagree; to 4 = strongly disagree. The positively worded items 4 and 6 were reverse coded. Total scores ranged from 7 to 28, with higher scores indicating higher levels of mastery.

#### 2.2.2. Caregiver Burden

Perceived burden was assessed using the Zarit burden interview (ZBI) [25]. The ZBI is one of the most widely used scales to assess burden experienced by caregivers and it has been validated in multiple settings. This study used the original 22-item ZBI. Each item was scored on a 5-point scale: 0 = never; 1 = rarely; 2 = sometimes; 3 = quite frequently; to 4 = nearly always. A higher score indicates higher burden, and the total score ranges from 0 to 88.

#### 2.2.3. Depression

Depressive symptoms were measured by the depression subscale of the hospital anxiety and depression scale (HADS-D), which has been shown to reliably assess depressive symptoms in clinical and general populations. HADS-D contains seven items rated on a four-point scale from 0 to 3 [26]. The total score ranges from 0 to 21, with a higher score indicating higher levels of depression [27].

#### 2.2.4. Health-Related Quality of Life

Health related quality of life (HRQoL) was measured using the short form-12 version 2 (SF-12v2). The SF12v2 is widely used and well validated in a variety of population groups as a quality of life measure. The SF-12v2 comprises 12 items which can be aggregated into a physical component summary (PCS) and a mental component summary (MCS), which measure physical and mental HRQoL, respectively [28]. The component scores were scored using RAND software and ranged from 0 to 100. Lower scores on the PCS or MCS, respectively, represent a higher likelihood and severity of physical and mental health conditions.

#### 2.2.5. Demographics Characteristics

Caregiver demographics including age, gender, education level, patient-caregiver relationship, living arrangements, and presence of domestic helpers were also collected.

### 2.3. Data Analysis

Analyses were performed using IBM SPSS Statistics for Windows, version 25.0 [29] and the R project for statistical computing (R-3.6.3) [30] with ltm [31] and lavaan [32] packages. Descriptive statistics were used to summarize the sample characteristics.

Confirmatory factor analysis (CFA) was conducted to assess whether the scale conformed to the one-factor model. As the mastery scale responses were treated as categorical data, the robust weighted least squares and variance adjusted (WLSMV) estimator was used. Acceptable model fit for CFA was assessed by a combination of indices (with the cut-off criteria within brackets): comparative fit index (CFI ≥ 0.95), and Tucker–Lewis index (TLI > 0.90), root mean square error of approximation (RMSEA < 0.08), and standardized root mean square residual (SRMR ≤ 0.08) [33,34,35].

After testing the CFA model, Samejima’s graded response model (GRM) [36] was used to generate IRT parameters for item response category characteristics curves (ICC) and item information curves (IIC). The ICC, with the *x*-axis representing the mastery level being measured and the *y*-axis representing the probability of endorsing a particular option to an item, describes the relationship between an individual’s mastery trait and their responses to an item. An item which performs well has a shape where the category thresholds are arranged in an ordered manner (e.g., peak of category 2 comes after category 1 and before category 3).

The IIC shows the level of sensitivity of a mastery item (*y*-axis) and how the sensitivity varies depending on whether individuals have below average, average, and above average mastery (*x*-axis) [37]. In general, an item in the mastery scale was considered for inclusion if the item offered (a) greater discrimination parameters (i.e., steeper slopes), (b) broader range of difficulty parameters spread across the full continuum of mastery trait, and (c) higher levels of item information—along the range of measured mastery trait [38].

Subsequently, the reliability and construct validity of the five-item mastery scale was also compared against the original seven-item mastery scale. We used McDonald’s omega (ω) as it provides better estimates of reliability than Cronbach’s alpha [39], especially when measuring a one-factor scale [40]. Construct validity was examined through CFA, Pearson’s correlation coefficients between the five-item and seven-item scale, and correlation coefficients between the two versions of scale and measures of caregiver depression, caregiver burden and HRQoL.

## 3. Results

### 3.1. Sample Characteristics

Of the 489 caregivers approached, 97 refused participation yielding a final sample of 392 caregivers. Their mean age was 57.6 years old (SD = 11.5). The majority were female (66.3%), Chinese (84.4%), children of the care-recipients (72.7%), and living with their care-recipients (78.3%). About 50% of the participants were working either full or part-time and had a live-in domestic helper. Their mastery levels covered the entire spectrum of possible mastery scores, ranging from 7 to 28 (mean = 19.5, SD = 3.3). The sample characteristics are presented in Table 1. In total, 174 of the 392 caregivers (44%) were caring for a person with dementia. All care-recipients required at least a one-person assistance with their activities of daily living.

### 3.2. Confirmatory Factor Analysis Statistics

Overall, results of the CFA of the seven-item mastery scale showed moderate fit for a one-factor structure (CFI = 0.93; TLI = 0.89; RMSEA = 0.07; SRMR = 0.05). Factor loadings for all items were significant (*p* < 0.001) and ranged from 0.32 to 0.75. Items 4 and 6 had the lowest factor loadings of 0.32 and 0.41, respectively, as compared to the other 4 items with factor loadings of ≥0.63.

### 3.3. IRT Parameters and Information

As seen in Table 2, the discrimination parameters ranged from 0.85 to 2.76. The difficulty parameters reflected a wide range of the mastery construct ranging from −5.05 to 2.18, indicating that the mastery scale was generally more useful in identifying individuals at the extreme low spectrum of mastery level.

As seen in the ICC (Figure 1), most items presented with shapes and category thresholds that were distinct and ordered, meaning that participants with low mastery levels tend to endorse the lowest category on the items. The exceptions were items 4 and 6 which did not have distinct peaks and had poor item discrimination parameters (Table 2). Item response curves also showed that one response option (“agree”) largely dominated the mastery trait continuum for these two items.

The IICs for each item in the mastery scale are shown in Figure 2. Item 5 generally has the highest item information across the continuum followed by item 1, while items 4 and 6 have low levels of item information with no peaks. Items 2 and 7 also appeared to have broader coverage of the continuum at very low levels of mastery.

### 3.4. Validity and Reliability of Mastery Scale

The CFA of the five-item mastery scale showed moderate fit to the data (CFI = 0.97; TLI = 0.94; RMSEA = 0.067; SRMR = 0.034). Factor loadings for all items were significant (*p* < 0.001) and ranged from 0.64 to 0.77.

Results of the Pearson correlations between both versions of the mastery scales and the caregiver outcomes are presented in Table 3. Total scores of the two versions of mastery scale were strongly correlated with one another (r = 0.96, *p* < 0.001 and had comparable reliability (7-item mastery: McDonald’s ω = 0.80, 5-item mastery: McDonald’s ω = 0.82). Notably, both five-item and seven-item versions of the mastery scale were negatively correlated with HADS-D and burden (r = −0.52 to −0.54, *p* < 0.001), and positively correlated with HRQoL (r = 0.30 to 0.40, *p* < 0.001).

## 4. Discussion

Our study examined psychometric properties of the Pearlin Mastery scale [1] amongst data from caregivers of care dependent older adults. The CFA model supported our hypothesis that there is only a single predominant source of influence on the seven-item mastery scale responses, but the IRT model highlighted a need to consider the removal of two positively worded items. Subsequent analysis of the composite raw scores of the five- and seven-item versions showed that they have acceptable reliability, and construct validity through positive correlations with quality of life and negative correlations with burden and depression scales.

As hypothesized, the two positively worded items (items four and six) in the seven-item scale did not perform well, prompting further analysis of the five-item version. CFA analysis of the seven-item scale showed that the two positively worded items had the lowest factor loadings (≤0.41). IRT results also revealed that the two positively worded items have the lowest discriminative ability and lowest item information across the range of mastery levels.

There was a moderate negative association between mastery and burden. This notable finding could be attributed to caregivers with higher levels of mastery perceiving caregiving challenges as less stressful as compared to those with lower levels of mastery [17,41]. In addition, we also found a positive association between mastery and HRQoL. Although this relationship is weak, it is nevertheless still notable as HRQoL can be influenced by a myriad of factors aside from mastery.

The poor performance of the two positively worded items mirrors past studies which reflect poorer model fit and reliability for the seven-item version as compared to the five-item version [9,12,13,14]. While the difference in reliability statistics between the five and seven-item scales were not as pronounced in our sample, our study concurs that the five-item scale demonstrates a better model fit than the original scale. Importantly, the results in this study provide evidence that the five-item version has acceptable reliability and validity that is comparable to that of the original seven-item version, which is consistent with the previous studies [9,12,13,14].

Mastery is a construct with positive connotations. Hence, it may raise some concerns when the original seven-item scale has only two positively worded items while the five-item version has no positively worded items. Here, it is important to consider the context in which the Pearlin Mastery scale was developed. It was initially developed to assess individuals in stressful roles [1] and subsequently incorporated by the scale developers into the stress process model for caregivers [2]. While the specifics of the scale development were not detailed in the literature, it is reasonable to conclude that there would be greater motivation to detect negative experiences as compared to positive ones. Thus, we believe that the content of the items is representative of the mastery construct even though the valence of the items may seem counter intuitive. Nonetheless, the five-item version of the mastery scale is well-suited for screening individuals.

After reviewing the items in the scale, no conclusive reasons were found to suggest that the positive items systematically differed from the negative ones (e.g., measuring something different than the other items or too complex for participants), leading to their poor fit. A possible reason for their poor performance could be wording effects which affect these reverse-worded items, leading to the measurement of an unrelated artefact that was not intended to be measured [42,43,44,45,46,47]. This was also suggested by Chen, Hsiung, Chung, Chen, and Pan [12] in their own analysis of the mastery scale. Future studies can consider adding new positively worded items to the scale to understand if the poor performance of the positive items is related to qualitative aspects of the items, or if this is purely a method effect.

Our work extends the use of Pearlin mastery tool beyond community dwelling individuals and patient populations [1,8,9,10,11,12] to family caregivers of care dependent older adults. This is important as the tool can be used in the clinical and research settings to determine the mastery level of the caregivers. A high level of mastery in caregivers can help mitigate the stress and negative health consequences faced by caregivers [15]. Furthermore, caregivers with high level of mastery would more likely engage in positive thinking and coping and utilize problem-solving behaviors in managing their caregiving situations [3].

### Limitations and Future Directions

Our study was conducted amongst family caregivers of hospitalized older adults. Hence, our findings may be more applicable to this sample of caregivers. Furthermore, hospitalization of one’s care-recipient may be a stressful period for caregivers. Nevertheless, mastery is a rather stable construct and may not be affected by the setting which it is measured in [48]. Future studies can further investigate the performance of the seven-item and five-item scale amongst caregivers in another setting. In addition, the criterion validity, known-group validity, and stability of the five-item mastery scale that have not been tested in this study can be examined in future studies.

## 5. Conclusions

The seven-item Pearlin’s Mastery scale was tested to be a valid and reliable tool to be used among the caregiver population, particularly in assessing family caregivers of care dependent older adults in the Singapore setting. This study has demonstrated evidence of a brief five-item scale that maintains the reliability and validity of the original scale, which will be valuable for academic research and practical survey implementation.

## Figures and Tables

**Figure 1 ijerph-19-04639-f001:**
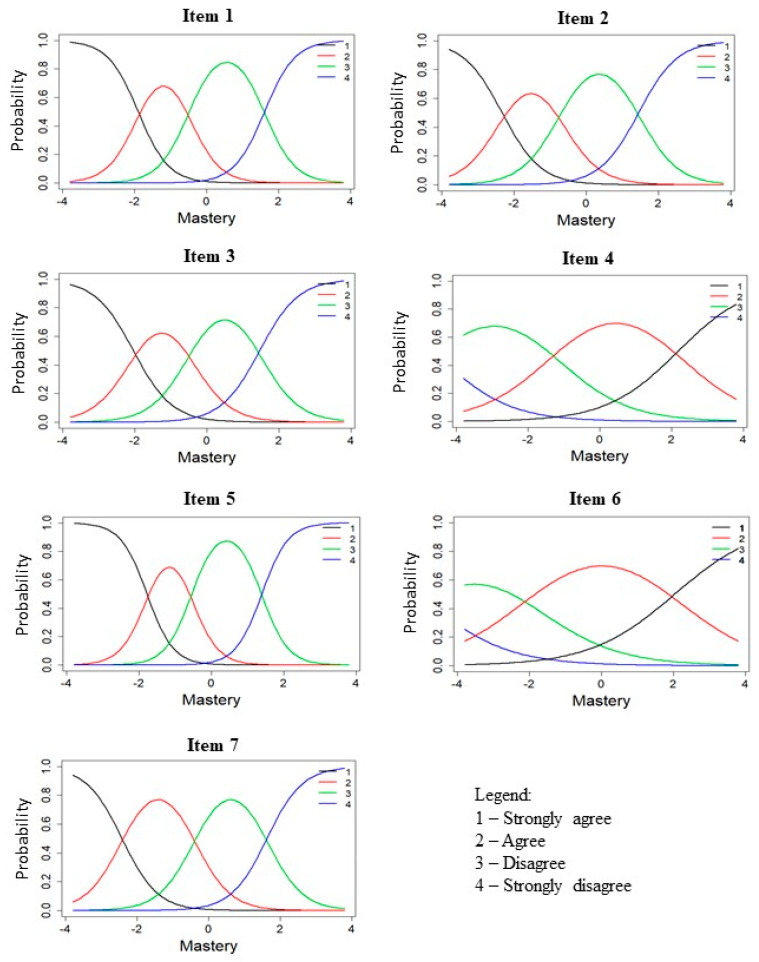
Item response category characteristic curves for mastery items.

**Figure 2 ijerph-19-04639-f002:**
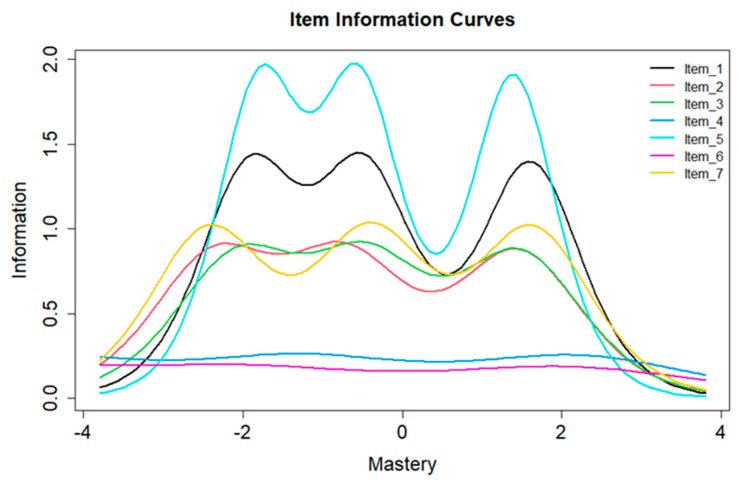
Item information curves for mastery items.

**Table 1 ijerph-19-04639-t001:** Demographics characteristics of participants (n = 392).

Demographic Characteristics	n (%)/Mean ± SD
Age, year	57.6 ± 11.5
Female	260 (66.3)
Married	217 (55.4)
Ethnicity	
Chinese	331 (84.4)
Malay	27 (6.9)
Indian	23 (5.9)
Others	11 (2.8)
Working (Full/Part-time)	200 (51)
Live-in domestic helper	194 (49.4)
Relationship to care-recipient	
Spouse	61 (15.6)
Child	285 (72.7)
Others	46 (11.7)
Living with care-recipient	307 (78.3)
Mastery (range 7 to 28)	19.5 ± 3.3
ZBI Burden (range 0 to 88)	28.5 ± 15.1
Depression, HADS subscale (range 0 to 21)	5.6 ± 4.3
Physical component summary, SF-12 subscale (range 0 to 50)	47.3 ± 8.1
Mental component summary, SF-12 subscale (range 0 to 50)	45.4 ± 8.8

Notes: SD = standard deviation; ZBI = Zarit Burden Interview; HADS = Hospital Anxiety and Depression Scale; SF-12 = The 12-Item Short Form Health Survey version 2.

**Table 2 ijerph-19-04639-t002:** Descriptive statistics and graded response model item parameters.

Item	Graded Response Model Parameters			
	*a*	*b* _1_	*b* _2_	*b* _3_
1	2.36	−1.90	−0.50	1.61
2	1.86	−2.33	−0.73	1.45
3	1.86	−2.04	−0.47	1.46
4	1.00	−4.61	−1.29	2.18
5	2.76	−1.78	−0.56	1.39
6	0.85	−5.05	−2.02	2.03
7	2.00	−2.43	−0.40	1.63

Notes: ***a*** = discrimination; ***b*_1_** = difficulty parameter (participants endorsing 1 versus ≥2); ***b*_2_** = difficulty parameter (participants endorsing 2 versus ≥3); ***b*_3_** = difficulty parameter (participants endorsing 3 versus 4).

**Table 3 ijerph-19-04639-t003:** Reliability and Validity of Original Mastery and Shortened Mastery scales.

Scale	Correlation with Original Scale	McDonald’s ω	Depressive Symptoms	Caregiver Burden	Health Related QoL	
					Physical	Mental
Mastery 7-item	-	0.80	−0.53 *	−0.52 *	0.32 *	0.40 *
Mastery 5-item	0.96 *	0.82	−0.52 *	−0.54 *	0.30 *	0.40 *

Notes: Mastery 5-item = items 1, 2, 3, 5 & 7; * *p* < 0.001.

## Data Availability

Data not available due to ethical restrictions.

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
