# Peer review of "Psychometrics of the Pearlin Mastery Scale among Family Caregivers of Older Adults Who Require Assistance in Activities of Daily Living"

_ijerph, 2022, doi:10.3390/ijerph19084639_

Round 1
Reviewer 1 Report
This paper presents results of a psychometric analysis of the Pearlin Master Scale for family caregivers of older adults. Overall, this is a well-written psychometric paper that provides support for use of this scale in a family caregiver population.
The rationale for the study is well-supported in the Introduction. Two minor comments:
- Line 44: the negative health effects of what?
- Line 45: awkward wording – perhaps replace “can mediate between” to “mediates the relationship between”?
Methods
- Line 82: should be caregivers of patients or caregivers whose patients (not caregivers of whose patients)
- How was inclusion criterion #2 (requiring assistance in ADLs) determined?
- Line 107: remove “of” so it reads “comprises 12 items”
- Are all other measures validated in Chinese as well?
Results
- How many persons were approached to participate to obtain the 392 who did participate?
- Overall results are clearly presented and the Figures are incredibly helpful.
- Is there any summary of how much assistance patients required?
- It would be helpful to note how many took the scale in Chinese vs English (in Table 1).
Discussion
- Very balanced discussion that nicely addresses the potential issues. More specific suggestions for how to reword or rewrite positively framed items would be helpful.
Author Response
Dear Reviewer 1,
Thank you for providing us the opportunity to revise our manuscript. We are very grateful for your insightful and important feedback. We have carefully considered each of them and our point-by-point responses to your feedback can be found below. Your feedback has been incorporated into our revised manuscript using track changes. We hope they adequately addressed your concerns.
Thank you.
Reviewer 1
This paper presents results of a psychometric analysis of the Pearlin Master Scale for family caregivers of older adults. Overall, this is a well-written psychometric paper that provides support for use of this scale in a family caregiver population.
The rationale for the study is well-supported in the Introduction. Two minor comments:
- Line 44: the negative health effects of what?
Response: We have edited line. It now reads “negative health effects due to stress associated with caregiving”.
- Line 45: awkward wording – perhaps replace “can mediate between” to “mediates the relationship between”?
Response: We have edited line 45 to reflect the suggested edits.
Methods
- Line 82: should be caregivers of patients or caregivers whose patients (not caregivers of whose patients)
Response: We have edited line 82 accordingly.
- How was inclusion criterion #2 (requiring assistance in ADLs) determined?
Response: In the revised manuscript we added that “The inclusion criteria was determined via patient’ medical records or asking the nurse-in-charge of the patient.” (line 84)
- Line 107: remove “of” so it reads “comprises 12 items”
Response: We have edited this line to reflect the suggested edits (line 109)
- Are all other measures validated in Chinese as well?
Response: Yes. We have included a line with citation to reflect this: “Chinese versions of the scales used were previously validated” (line 90)
Results
- How many persons were approached to participate to obtain the 392 who did participate?
Response: We added that 489 caregivers were approached, and 97 refused participation yielding a final sample of 392 caregivers.” (see line 159)
- Overall results are clearly presented and the Figures are incredibly helpful.
- Is there any summary of how much assistance patients required?
Response: Generally patients required at least one-man assistance with their activities of daily living. We added this information to the text in the results section. We also added that percentage of caregivers who were caring for a person with dementia. (please see lines 265-267)
- It would be helpful to note how many took the scale in Chinese vs English (in Table 1). Response: 32%lf of the participants took the scale in Chinese. There was no significance difference in the outcome between those who took the scale in Chinese vs English.
- Discussion
- Very balanced discussion that nicely addresses the potential issues. More specific suggestions for how to reword or rewrite positively framed items would be helpful.
- Response: We thank the Reviewer for this feedback. We agree the Reviewer and think that this will be an interesting area for future study to re-examine how to reword the items to be positively frame along with their influence on the psychometric properties of the tool.

Reviewer 2 Report
This manuscript evaluates the psychometric properties of the Pearlin Mastery Scale among caregivers. Using data from 392 caregivers of older adults hospitalized in a tertiary care facility, the authors used confirmatory factor analysis and item response theory to assess the construct validity, including convergent validity, of the 7- and 5-item versions of the mastery scale. The paper fills an important gap in the literature and is very well written. The methods are appropriate and clearly described, and the results are interpreted accurately. I have only a few small suggestions:
- Methods 2.1: Please indicate the response rate among caregivers, i.e., how many caregivers were approached to reach the sample size of 392?
- Methods 2.2: When describing the depression and HRQOL measures, please add some information about their validity/reliability (i.e., a short statement with a citation).
- Results: You might reiterate here that the participants were caregivers – either in the first sentence and/or in Table 1.
- Results: Did you collect any information about the health condition(s) of the care recipients? Not critical, but if available it would be nice to know what types of caregivers these participants represent.
- Results 3.2: I believe the last sentence should be >= 0.63 (rather than less than or equal to).
- Table 1: Are the ranges reported the observed ranges or just the possible ranges of the scales? Please include the observed range, if this is not already done.
- Results 3.4: In the second paragraph, you might switch the order so the text matches the order of Table 3.
- Results 3.4/Discussion: While the text is accurate, I consider correlations in the 0.3-0.4 range to be weak, so you might mention this explicitly in the discussion. I wouldn’t expect HRQOL to be driven by mastery so these observed correlations are still quite notable. The same is true for burden – while 0.5 is a moderate association it is noteworthy that mastery has this strong a correlation with burden. If there is space, you may add some comment on this.
Author Response
Dear Reviewer 2,
Thank you for providing us the opportunity to revise our manuscript. We are very grateful for your insightful and important feedback. We have carefully considered each of them and our point-by-point responses to your feedback can be found below. Your feedback has been incorporated into our revised manuscript using track changes. We hope they adequately addressed your concerns.
Reviewer 2
This manuscript evaluates the psychometric properties of the Pearlin Mastery Scale among caregivers. Using data from 392 caregivers of older adults hospitalized in a tertiary care facility, the authors used confirmatory factor analysis and item response theory to assess the construct validity, including convergent validity, of the 7- and 5-item versions of the mastery scale. The paper fills an important gap in the literature and is very well written. The methods are appropriate and clearly described, and the results are interpreted accurately. I have only a few small suggestions:
- Methods 2.1: Please indicate the response rate among caregivers, i.e., how many caregivers were approached to reach the sample size of 392?
Response: We have included a line to reflect this information. “Of the 489 caregivers approached, 97 refused participation yielding a final sample of 392 caregivers.” (line 159)
- Methods 2.2: When describing the depression and HRQOL measures, please add some information about their validity/reliability (i.e., a short statement with a citation).
Response: We have incorporated the suggested edits to include a short statement with citation. (line 108 and 115)
- Results: You might reiterate here that the participants were caregivers – either in the first sentence and/or in Table 1.
Response: We have edited the first sentence to reflect this: “a final sample of 392 caregivers.” (line 159)
- Results: Did you collect any information about the health condition(s) of the care recipients? Not critical, but if available it would be nice to know what types of caregivers these participants represent.
Response: 174 of the 392 caregivers (44%) were caring for a person with dementia. In addition, their care-recipients required either one- or two-man assistance with their activities of daily living. (Please see Results section, line 85)
- Results 3.2: I believe the last sentence should be >= 0.63 (rather than less than or equal to).
Response: Thank you for spotting the typo. We have edited it to reflect more than or equal to ”≥".
- Table 1: Are the ranges reported the observed ranges or just the possible ranges of the scales? Please include the observed range, if this is not already done.
Response: The range is intended to reflect the possible range of the scale. In our sample, the observed range is the same.
- Results 3.4: In the second paragraph, you might switch the order so the text matches the order of Table 3.
Response: We have edited this section to reflect the suggested edits (Results 3.4)
- Results 3.4/Discussion: While the text is accurate, I consider correlations in the 0.3-0.4 range to be weak, so you might mention this explicitly in the discussion. I wouldn’t expect HRQOL to be driven by mastery so these observed correlations are still quite notable. The same is true for burden – while 0.5 is a moderate association it is noteworthy that mastery has this strong a correlation with burden. If there is space, you may add some comment on this.
Response: Thank you. We have included a paragraph to comment on this portion of the results. (page 8, lines 282 to 286)
